# Differential Effects of Somatostatin, Octreotide, and Lanreotide on Neuroendocrine Differentiation and Proliferation in Established and Primary NET Cell Lines: Possible Crosstalk with TGF-β Signaling

**DOI:** 10.3390/ijms232415868

**Published:** 2022-12-14

**Authors:** Hendrik Ungefroren, Axel Künstner, Hauke Busch, Sören Franzenburg, Kim Luley, Fabrice Viol, Jörg Schrader, Björn Konukiewitz, Ulrich F. Wellner, Sebastian M. Meyhöfer, Tobias Keck, Jens-Uwe Marquardt, Hendrik Lehnert

**Affiliations:** 1First Department of Medicine, University Hospital Schleswig-Holstein (UKSH), Campus Lübeck, D-23538 Lübeck, Germany; 2Institute of Pathology, University Hospital Schleswig-Holstein (UKSH), Campus Kiel, D-24105 Kiel, Germany; 3Medical Systems Biology Group, Lübeck Institute of Experimental Dermatology, University of Lübeck, D-23538 Lübeck, Germany; 4Institute for Cardiogenetics, University of Lübeck, D-23538 Lübeck, Germany; 5Institute for Clinical Molecular Biology, University of Kiel, D-24118 Kiel, Germany; 6Clinic of Oncology, University Hospital Schleswig-Holstein (UKSH), Campus Lübeck, D-23538 Lübeck, Germany; 7Medical Clinic and Policlinic, University Hospital Hamburg-Eppendorf, D-20251 Hamburg, Germany; 8Department of Surgery, University Hospital Schleswig-Holstein (UKSH), Campus Lübeck, D-23538 Lübeck, Germany; 9Institute of Endocrinology and Diabetes, University of Lübeck, D-23538 Lübeck, Germany; 10German Center of Diabetes Research, D-85764 Neuherberg, Germany; 11University of Salzburg, 5020 Salzburg, Austria

**Keywords:** gastroenteropancreatic neuroendocrine tumor (GEP-NET), BON-1 (BON), QGP-1 (QGP), NT-3, octreotide (OCT), lanreotide (LAN), microRNA (miRNA), somatostatin (SST), somatostatin analogues (SSAs)

## Abstract

GEP-NETs are heterogeneous tumors originating from the pancreas (panNET) or the intestinal tract. Only a few patients with NETs are amenable to curative tumor resection, and for most patients, only palliative treatments to successfully control the disease or manage symptoms remain, such as with synthetic somatostatin (SST) analogs (SSAs), such as octreotide (OCT) or lanreotide (LAN). However, even cells expressing low levels of SST receptors (SSTRs) may exhibit significant responses to OCT, which suggests the possibility that SSAs signal through alternative mechanisms, e.g., transforming growth factor (TGF)-β. This signaling mode has been demonstrated in the established panNET line BON but not yet in other permanent (i.e., QGP) or primary (i.e., NT-3) panNET-derived cells. Here, we performed qPCR, immunoblot analyses, and cell counting assays to assess the effects of SST, OCT, LAN, and TGF-β1 on neuroendocrine marker expression and cell proliferation in NT-3, QGP, and BON cells. SST and SSAs were found to regulate a set of neuroendocrine genes in all three cell lines, with the effects of SST, mainly LAN, often differing from those of OCT. However, unlike NT-3 cells, BON cells failed to respond to OCT with growth arrest but paradoxically exhibited a growth-stimulatory effect after treatment with LAN. As previously shown for BON, NT-3 cells responded to TGF-β1 treatment with induction of expression of *SST* and *SSTR2*/*5*. Of note, the ability of NT-3 cells to respond to TGF-β1 with upregulation of the established TGF-β target gene *SERPINE1* depended on cellular adherence to a collagen-coated matrix. Moreover, when applied to NT-3 cells for an extended period, i.e., 14 days, TGF-β1 induced growth suppression as shown earlier for BON cells. Finally, next-generation sequencing-based identification of microRNAs (miRNAs) in BON and NT-3 revealed that SST and OCT impact positively or negatively on the regulation of specific miRNAs. Our results suggest that primary panNET cells, such as NT-3, respond similarly as BON cells to SST, SSA, and TGF-β treatment and thus provide circumstantial evidence that crosstalk of SST and TGF-β signaling is not confined to BON cells but is a general feature of panNETs.

## 1. Introduction

Gastroenteropancreatic neuroendocrine tumors (GEP-NETs) are heterogeneous tumors with respect to clinical and biological features that originate from the pancreas or the intestinal tract [1]. Some GEP-NETs exhibit very slow growth, and most cases are diagnosed at an advanced stage when options for curative treatments are not available anymore. While some tumors grow rapidly and do not evoke symptoms, others cause hormone hypersecretion and associated symptoms [1]. While only a few patients with NETs are amenable to curative tumor resection, for many patients, only palliative treatments remain, such as with somatostatin (SST) or SST analogs (SSAs), such as octreotide (OCT) or lanreotide (LAN) [2]. GEP-NET standard therapy with SSAs for symptomatic treatments is based on the fact that most GEP-NETs overexpress receptors for SSTs, i.e., somatostatin receptor 2 (SSTR2), which inhibit the release of numerous hormones and other secretory proteins [2,3]. Besides symptomatic management, recent research has demonstrated that SSAs display antiproliferative effects and inhibit tumor growth through SSTR2. Long-acting formulations of SSAs keep tumor growth under control over long periods, as revealed by multiple clinical trials demonstrating high rates of disease stabilization upon treatment with SSAs, and are state-of-the-art treatment options in SSTR expressing G1 and G2 tumors according to current guidelines [4].

The two available pancreatic NET cell lines, BON-1 (BON) [5] and QGP-1 (QGP) [6], have so far been widely used as a bona fide NET model. However, several characteristics, such as the fast proliferation rates/high doubling times; the low expression of NE markers in these tumor cell lines; and the recent discovery of mutations in *NRAS* (in BON), *KRAS* (in QGP), and *TP53* (in both lines) [7,8] added to these concerns since well-differentiated NETs usually do not harbor such mutations and have questioned their usefulness as NET models. Given that both cell lines have been used for more than two decades and proliferate much faster than in their early passage numbers, the cells may have acquired a more malignant phenotype in culture. The novel pancreatic NET cell line NT-3 has been characterized by us previously and shown to resemble the patient’s original tumor closely: it has a well-differentiated phenotype, is functionally active, and has a slow growth rate in vitro and in vivo NT-3 cells thereby recapitulate the cardinal features of NETs and provide a highly relevant but hitherto unavailable preclinical NET model. In addition, NT-3 cells lack mutations in *RAS* or *TP53,* thereby resembling the non-mutated status of well-differentiated NET for these oncogenes [9]. NT-3 and BON, QGP cells have been characterized by us in a follow-up study and shown to display a NE and an epithelial phenotype [10].

Moreover, both BON and QGP resemble immature/non-functional pancreatic β/δ-cells or pancreatic endocrine progenitors, while NT-3 cells are similar to mature functional β-cells [10]. According to a recently proposed classification of transcriptional subtypes in panNETs [11], NT-3 cells resemble the “islet/insulinoma tumors” (IT) subtype. In contrast, BON and QGP cells were tentatively classified as “metastasis-like/primary” (MLP).

OCT is a widely used synthetic SSA that acts through SSTRs and significantly improves the management of NETs. However, the molecular mechanisms leading to successful disease control or symptom management, especially when SSTR levels are low, are largely unknown. Even cells or tissues expressing low levels of SSTRs show significant responses to OCT, suggesting that OCT may signal through alternative mechanisms still requiring the expression of SSTRs but also possibly involving the participation of different genes and a novel signaling framework. Along the same lines are earlier findings in which the SST signaling pathway has been shown to mediate the effect of TGF-β on cell proliferation and differentiation in BON cells. More specifically, SST signaling determines the growth-inhibitory response of BON cells to TGF-β. TGF-β induces the production of SST and potentially activates a negative growth autocrine loop of SST, which leads to the downstream induction of growth-inhibitory effectors, i.e., p21^Waf1/Cip1^ and p27^Kip1^, as well as downregulation of c-Myc [12]. However, while the first study only used human midgut carcinoid CNDT2.5 cells, the latter employed only BON cells.

Nevertheless, both studies failed to evaluate the response of this growth factor in other panNET cell lines of either established, i.e., QGP, or, more importantly, primary origin, i.e., NT-3. Furthermore, these studies did not analyze SST or SSAs other than OCT, such as LAN, or cellular responses other than proliferation, e.g., NE differentiation. Moreover, the crosstalk of the SST-SSTR2/5 and TGF-β1 signaling system has been studied only in BON cells resembling immature β-cells/pancreatic progenitors but not in mature β-cells such as NT-3; we sought to know whether SST-TGF-β co-signaling also operates in functional β-cells.

In addition, since many studies have focused on the effects of SST and SSAs on proliferation, we also studied whether SST, OCT, or LAN treatment also impacts the NE phenotype. Since the expression of NE markers such as CgA, SYP, SST, SSTR2, and SSTR5 is generally low in BON and QGP cells, it was necessary to primarily employ qPCR analysis for an accurate quantitative assessment of their expression. Indeed, OCT has been shown to act mainly through changes in gene expression [13].

Here, we compared, for the first time systematically, the differentiation and proliferation response of the three NET cell lines, BON, QGP, and NT-3, to treatment with SST; the two SSAs, OCT and LAN; and TGF-β1. We particularly focused on the underinvestigated signaling mode of the SSAs through TGF-β. Since a previous next-generation sequencing (NGS)-based identification of microRNAs (miRNAs) in BON, QGP, and NT-3 cells revealed the presence of miRNAs involved in β-cell function and differentiation [10], we asked, in addition, which of these miRNAs may be subject to regulation by SST or SSAs.

## 2. Results

### 2.1. Regulation of NE Markers by SST or SSAs in Established and Primary NET Cell Lines

Here, we studied by qPCR and immunoblot analysis whether the NE phenotype defined by the above markers is altered by treatment with SST, OCT, or LAN (each tested at 10 nM and 1 µM concentrations). To this end, in BON cells treated with either concentration of SST, OCT, or LAN, the mRNAs for *CHGA*, *SYP*, *SSTR2*, and *SSTR5* were found to be upregulated (Figure 1A). In BON cells, we also observed an induction of *SLC2A2* by 1 µM OCT and both concentrations of LAN (Figure 1A). Moreover, in BON, we evaluated the response of the early endocrine progenitor and islet cell marker gene *NEUROD1* to challenge with SST, OCT, or LAN. All three compounds induced *NEUROD1* at both concentrations, except for 1 µM SST (Figure 1A, top right panel).

In QGP cells, *CHGA* was downregulated by 1 µM SST and both concentrations of OCT and LAN (Figure 1B). However, all three agents stimulated *SLC2A2* and *SSTR2*, while *SSTR5* was only induced by 1 µM SST or 1 µM LAN (Figure 1B). No statistically significant effects of either of the three SSAs were seen for *SYP* (Figure 1B).

NT-3 cells cultured under semi-adherent conditions exhibited a slight decrease in *CHGA* mRNA levels relative to vehicle control only after treatment with 1 µM LAN (Figure 1C). In regulating *SLC2A2*, SST and OCT displayed somewhat antagonistic effects; while SST induced a 2.6-fold increase over vehicle controls, OCT reduced *SLC2A2* mRNA levels to 19.2% of controls and LAN was without an effect (Figure 1C). In contrast, mRNA levels of *SST* (when assessed as a target gene), *SSTR2,* and *SYP* failed to change significantly after stimulation with any of these agents (Figure 1C), suggesting regulation at the protein level. Of note, *SSTR5* mRNA was more abundant after treatment with OCT or LAN but not SST (Figure 1C).

Immunoblot analysis was then used to measure the protein levels of CgA, SYP, and SSTR2 in NT-3 and CgA and SYP in BON and QGP cells, in response to treatment with either SST, OCT, or LAN. We failed to detect CgA protein in QGP cells, however, in BON cells, CgA was readily detectable and induced by all three agents following densitometry-based quantification of band intensities (Figure 1D, top left). In NT-3 cells, the abundance of CgA protein appeared to be reduced in response to stimulation with all three agents; however, only for LAN, this effect was statistically significant (Figure 1D, top middle). Interestingly, the levels of SYP protein were increased by SST, OCT, or LAN in both BON (Figure 1D, top right) and QGP cells (Figure 1D, bottom left) but were refractory to regulation in NT-3 cells (Figure 1D, bottom middle) in accordance with the mRNA data (Figure 1C). SSTR2 protein could be detected only in NT-3, but following normalization for equal protein loading, no regulatory effect by any of the SSAs was noted (Figure 1D, bottom right). Taken together, SST and SSAs regulate various NE genes in BON, QGP, and NT-3 cells. Of note, SST and, in particular, LAN often differed in their effects from those of OCT.

### 2.2. Regulation of SERPINE1, SST, and SSTR Expression by TGF-β1 in BON, QGP, and NT-3 Cells

The role of TGF-β in regulating the SST or SSTR expression has only been studied in BON cells so far [12]. In order to evaluate sensitivity to TGF-β stimulation in QGP and NT-3 cells, we initially challenged both cell lines (and BON as control) with TGF-β1 and determined the mRNA levels of *SERPINE1* (encoding plasminogen activator-inhibitor 1), an established TGF-β target gene, by qPCR analysis. Both BON and QGP (Figure 2A) cells responded albeit weakly to a 24 h (but not 48 h) stimulation period with induction of *SERPINE1*. In contrast, NT-3 cells reacted with a much more robust and time-dependent upregulation, which peaked at 48 h of stimulation (Figure 2B, left-hand panel).

Unlike BON and QGP, NT-3 cells grow semi-adherently, with a fraction of cells adhering to the collagen surface. In contrast, others—particularly those involved in forming spheres—do not have direct contact with the collagen-coated surface [9]. Therefore, it was interesting to study if non-adherent cells would respond differently to TGF-β1 stimulation. To this end, when NT-3 cells were cultured under conditions that prevented their adhesion (by omitting the collagen coating of the culture flasks), basal expression levels dropped, and the stimulatory TGF-β1 effect on *SERPINE1* was completely abolished (Figure 2B, right-hand panel).

In NT-3 cells, we also measured whether TGF-β1 impacts the expression of *SST*, *SSTR2,* or *SSTR5*. Following stimulation with TGF-β1 for 24 h, we observed upregulation of *SST* (fold induction: 2.37 ± 0.38, *p* = 0.012, Wilcoxon test), *SSTR2* (2.3 ± 0.48, *p* = 0.028, Wilcoxon test) and *SSTR5* (4.26 ± 1.32, *p* = 0.046, Wilcoxon test) under adherent conditions (Figure 2C). The evaluation of additional time points, namely 48 h and 72 h of TGF-β1 treatment, revealed a time-dependent decline in *SST* expression (48 h: 1.92 ± 0.41-fold, *p* = 0.021; 72 h: 1.24 ± 0.12-fold, not significantly different to control), but not *SSTR2* expression (Figure 2C). Our results show that (i) all three cell lines are TGF-β sensitive, (ii) the ability to respond to this growth factor depends on cell–matrix interactions, and (iii) TGF-β1 can potentially induce SST/SSTR2/SSTR5 signaling in NT-3 cells.

### 2.3. Effect of SST, SSAs or TGF-β1 on Cell Proliferation of BON, QGP and NT-3 Cells

In order to study the effects of SST and SSAs on proliferative activities, we performed cell counting assays in BON and QGP cells. Following a 72 h treatment period, both SST and OCT failed to show a significant reduction in cell numbers on either cell line, in agreement with a previous finding of lack of an effect of OCT in QGP proliferation [9]. However, LAN unexpectedly increased cell numbers to 126.9 ± 3.2% (*p* = 0.02, n = 3, Wilcoxon test) of controls in BON but reduced them to 89.05 ± 5.46% (*p* = 0.028, n = 4, Wilcoxon test) of controls in QGP cells (Figure 3A). As a control, we treated the cells with the phosphatidylinositol 3-kinase (PI3K) inhibitor LY294002 since inhibition of PI3K/AKT signaling suppresses tumor cell proliferation and NE marker expression in GI carcinoid tumors [14]. This intervention brought cell numbers down to 24.5 ± 8.0% of control (*p* = 0.005, n = 3, Wilcoxon test) in BON cells and to 64.1 ± 13.5% (*p* = 0.035, Wilcoxon test) in QGP cells. Alternatively, cells were exposed to the MEK inhibitor U0126, which also resulted in growth arrest (28.4 ± 4.7% of control, *p* = 0.002, n = 3, Wilcoxon test) in BON, and 43.75 ± 9.5% of control (*p* = 0.047, n = 3, Wilcoxon test) in QGP cells (Figure 3A). The effect of OCT on NT-3 cell proliferation had been evaluated previously after a 5-day treatment with 100 nM OCT, which reduced cell counts by 34.8% [9].

Above, we observed that (i) OCT or LAN can induce SSTR5 and (ii) TGF-β1 can induce the expression of SST, SSTR2, and SSTR5 in NT-3 cells. This suggested the possibility that in NT-3—as previously shown in BON cells [12]—, SST and TGF-β signaling interact to mediate the effect of TGF-β on cell proliferation and differentiation. However, the response to TGF-β1 concerning proliferation has not yet been evaluated in NT-3 cells. By using cell counting assays, we assessed the effect of TGF-β1 on cell growth in NT-3 cells. In contrast to BON cells, which had a doubling time of only 1.5 days, NT-3 grew slowly with a doubling time of 10.9 days [9]. Due to this low proliferation rate, treatment periods of 7 and 14 d rather than 3 d for BON (see Figure 3B) were chosen. A statistically significant growth-inhibitory effect of TGF-β1 on NT-3 cells was noted after 14 days of treatment but not after 7 days of treatment (Figure 3B). We conclude that BON cells failed in our hands to respond to OCT with growth arrest (in contrast to a previous report [12]) but paradoxically exhibited a growth stimulatory effect after treatment with LAN. Moreover, the data reveal for the first time that TGF-β1 is growth-suppressive in NT-3 cells when applied for an extended period.

### 2.4. Regulation of miRNAs Expressed in BON, QGP, and NT-3 Cells by SST and SSA

Non-coding RNAs, such as miRNAs, are essential regulators of post-transcriptional control of gene expression in panNET [15,16,17]. Given the observed effects of the SSAs on various cellular functions of panNET cell lines, we speculated that specific miRNAs should also be subject to regulation by SST, OCT, or LAN. To this end, we treated the three cell lines with either SST or the two SSAs and performed a differential miRNA analysis. For BON cells, three miRNAs were identified as significantly different between treatments (*p* < 0.1). The miRNAs AC079951.1 and AP001273.1 were more strongly expressed in SST-treated cells compared to vehicle controls (Figure 4).

Additionally, miR-221 was more abundant in vehicle controls than in OCT-treated BON cells (Figure 4). In NT-3 cells, miRNA AL449212.1 was expressed at a higher level in SST-treated cells compared to untreated cells. For QGP cells, no differentially expressed miRNAs were identified (Figure 4).

Finally, we strived to validate the functional activity of the identified miRs. Since for AC079951.1, AP001273.1, and AL449212.1 specific target genes are not known, we focused on miR-221, which has been demonstrated to possess oncogenic functions [18]. In pancreatic cancer, tissue inhibitor of metalloproteinase 2 (TIMP2) was reported to be a direct functional target of miR-221/222 [18]. Therefore, we measured TIMP2 expression in OCT-treated BON cells by qPCR analysis and noted increased levels of its mRNA when compared to control cells (Appendix A). Together, these findings reveal that at least SST and OCT can impact the regulation of certain miRNAs in NET cells and that OCT treatment of BON cells resulted in altered expression of an established miR-221 target gene.

## 3. Discussion

The primary goal of the present study was to assess the effects of SST, the SSAs, OCT, LAN, and TGF-β1 on NE differentiation, growth, and miRNA expression in primary (NT-3) and established (BON, QGP) panNET cell lines. As a secondary goal, we aimed to analyze the crosstalk between SSTR and TGF-β signaling. In a previous study with BON, QGP, and NT-3 cells, we evaluated basal levels of expression of NE markers such as CGA, GLUT2 (encoded by *SLC2A2*), SYP, SSTR2, and SSTR5 by both qPCR and immunoblotting [10]. These results confirmed the NE phenotype of BON and QGP cells, albeit SSTR2 and 5 expressions in BON and QGP were either weak or undetectable. In contrast, NT-3 cells presented with high mRNA expression of SSTR2 and SSTR5 (besides SSTR1 and SSTR3), and Western blot analysis revealed that the most clinically relevant SSTR2 was exclusively expressed in NT-3. At the same time, SSTR5 was also described in BON [9]. Here, in BON cells treated for 24 h with either SST or LAN, but not OCT, the mRNA for *CHGA* was found to be upregulated, and this stimulatory effect of SST, LAN (and OCT) was also observed at the protein level. Moreover, in BON cells, we observed the induction of *SLC2A2*, *SSTR2,* and *SSTR5* by OCT. *SYP* was upregulated by SST and OCT, while the immunoblots revealed upregulation by all three SSTs. In QGP cells, *CHGA* was downregulated by SST but remained refractory to regulation by OCT or LAN. However, all three agents stimulated *SLC2A2* in QGP, while *SYP* was induced only by LAN (unlike SYP protein, which increased in response to treatment with all three SSTs). Moreover, in QGP, *SSTR2* was induced by OCT and LAN, and *SSTR5* was induced by SST and LAN.

NT-3 cells cultured under adherent conditions exhibited a slight decrease in *CHGA* mRNA and CgA protein levels relative to vehicle controls only after treatment with LAN. Concerning the regulation of *SLC2A2*, SST, OCT, and LAN displayed divergent effects in NT-3; SST induced an increase over vehicle controls; OCT reduced *SLC2A2* mRNA levels; and LAN was without an effect. In contrast, mRNA levels of *SYP*, *SST* (when assessed as a target gene), and *SSTR2* failed to change significantly after stimulation with any of the three agents, which was reflected for SYP and SSTR2 also at the protein level. However, the abundance of *SSTR5* mRNA was slightly induced by both OCT and LAN. Thus, SST, OCT, and LAN display quite divergent effects on the NE markers when applied to the same cell line, but each agent also differs in their effects on these markers among the three lines. Our results nevertheless suggest that the mainly stimulatory effects of SST/OCT/LAN on NE marker expression, particularly CgA, SYP, SSTR2, and SSTR5, may reflect an increase in differentiation, which might be associated with a decrease in malignancy. Hence, the concept of these SSAs acting as pro-differentiation agents represents a previously underappreciated mode of anti-neoplastic function. Moreover, the transcriptional induction by OCT and LAN of their receptors, particularly SSTR5, may serve as a feed-forward loop to potentiate SST signaling, NE differentiation, or even the antiproliferative function of TGF-β (see below). Due to the amplification effect, even small changes in SSTR levels may be physiologically meaningful, eventually leading to a reduced malignant behavior of the tumor cells.

In order to study the impact of TGF-β on NT-3 and QGP cells, we evaluated the sensitivity of NT-3 and QGP cells to TGF-β stimulation using induction of *SERPINE1* as readout. Here, we observed that in NT-3 cells, TGF-β1 induced an increase in mRNA abundance of SERPINE1 and *SST*, *SSTR2,* and *SSTR5*, suggesting transcriptional upregulation as the underlying mechanism. In BON cells, the SST signaling pathway is a determinant in responding to TGF-β as a growth inhibitor [12]. These authors showed that adding either 10 ng/mL TGF-β or 100 nM OCT in fetal bovine serum (FBS)-free cell growth media independently exerted growth suppression on BON cells. They further demonstrated that both TGF-β-dependent growth inhibition and differentiation are mediated by SST signaling and that TGF-β upregulates *SST* and *SSTR2* to activate a negative growth autocrine loop [12]; SSTR2 on the membrane surface was inducible by both TGF-β (2-fold increase) and OCT (2.5-fold increase), in agreement with the OCT-induced induction of *SSTR2* mRNA in BON (Figure 1B) and NT-3 (Figure 2D) cells. Among the four receptor subtypes, only the human SSTR2 had a pronounced increase in transcriptional response and membrane surface representation stimulated by either TGF-β or OCT. This led the authors to suggest that SSTR2 represents the primary receptor subtype that mediates TGF-β-induced SST growth inhibition. Leu and coworkers also reported that a failure to activate the SST signaling pathway in BON cells resulted in a significant decrease in CgA, which could be rescued by adding OCT or TGF-β to activate the SST signaling pathway. The stronger TGF-β sensitivity of NT-3 vs. BON cells and the induction of SST, SSTR2, and SSTR5 by TGF-β1 in NT-3 indicates that crosstalk of SST-SSTR and TGF-β signaling is not confined to BON and hence immature, non-functional β-cells but appears to be a characteristic feature of both immature and mature /functional NET cells. Thus, although not explicitly tested in our study, the TGF-β-induced SST upregulation of SSTR2 described in BON may also operate in NT-3 cells, although direct treatment of NT-3 cells with SST peptide for 24 h failed to upregulate *SSTR2* (Figure 1C). Nevertheless, these cells may provide a model superior to BON in studying SST-TGF-β signaling crosstalk.

NT-3 cells grow semi-adherently, with a fraction of cells adhering to the collagen-coated culture flask surface. In contrast, others, particularly those involved in forming spheres, do not have direct contact with it. Of note, in NT-3 cells cultured under conditions that prevented their adhesion, the TGF-β1 effect on *SERPINE1* was wholly abolished (Figure 2C). This finding is consistent with earlier observations in the pancreatic ductal adenocarcinoma-derived cell line PANC-1 [19]. It may be related to the requirement for integrin ligation and activation of the small GTPase RAC1 (acting as part of the ROS-generating enzyme NADP(H) oxidase) for successful TGF-β signaling to matrix genes, i.e., *BGN* [19].

Since data on the effects of SSTs on cell proliferation of GEP-NET cell lines are scarce and, in part, contradictory, we set out to analyze the effects of SSTs on the proliferative activity of BON and QGP cells. We observed that OCT or SST (each at 1 µM) treatment alone for 3 d did not negatively impact BON and QGP cell proliferation as determined by cell counting. However, we unexpectedly observed a statistically significant increase in BON cell numbers upon treatment with LAN, suggesting that LAN can have a growth-stimulatory effect in BON cells. In contrast to BON and QGP, NT-3 cells, when treated for 5 d with 0.1 µM OCT, exhibited a 35% reduction in cell numbers [9]. Hence, given both the growth-arresting and *SST*/*SSTR2*/*SSTR5*-inducing effects of TGF-β1 in NT-3 (Figure 3 and Figure 2, respectively), SST/SSTR2 signaling may be involved in TGF-β-dependent growth inhibition as demonstrated previously in BON.

It should be mentioned that other studies could not find the effects of SSAs, especially OCT, treatment in BON and QGP cells when systematically analyzing cell proliferation or cell cycle distribution [20]. Likewise, in orthotopic BON tumors, treatment with LAN failed to inhibit tumor growth [21]. It has been speculated that this OCT resistance is due to low OCT binding and low SSTR, specifically SSTR2, expression compared to levels found in human NETs. In agreement with the study of Exner and colleagues [20] but in contrast to that of Leu and coworkers [12], we found that OCT (and SST) did not affect the growth of BON (Figure 3). This discrepancy may have been caused by the presence of serum in our assays vs. serum-free conditions in the Leu study. It is believed that the FBS content of 10% may have already activated the SST signaling pathway to a maximal extent. However, in vivo studies showed that another SSA, SMS 201-995, significantly inhibited the growth of BON tumors xenotransplanted into athymic nude mice but, interestingly, failed to inhibit the growth of BON cells in vitro [22], the latter observation being in agreement with our data on SST and OCT in BON and QGP cells. However, much like LAN in BON cells (in our study), SMS 201-995 exhibited a dose-dependent stimulation of growth on two other cell lines (A431 and KB human epidermoid carcinoma cells) in vitro [22], an effect that may have been mediated through the reduction in cyclic AMP production [22]. It remains to be seen whether such a mechanism is also involved in the growth-promoting LAN effect on BON cells. Nevertheless, our results support earlier evidence for the lack of authentic, tumor-like SSTR expression and growth-regulating functions in BON and QGP cells and point to the need for more physiologic tumor model systems [20]. Indeed, NT-3 cells may be a more suitable tumor cell model than BON and QGP, which is also supported by their higher physiologic levels of SSTR2 and SSTR5.

Despite low-level expression of SSTR2/5 in BON and QGP, these cells nevertheless displayed significant responses to OCT and LAN in our study. This suggests the possibility that these SSAs signal through alternative mechanisms, which—although these require an expression of SSTRs—may involve the participation of different genes and novel signaling networks. For instance, an effect of SSAs on tumor cell proliferation or differentiation may proceed through indirect effects and interactions with other proliferation-associated pathways. Su-Chen Li and colleagues first proposed that OCT may signal its effects through SSTRs by activating a set of genes that have not previously been associated with the conventional OCT signaling pathway. In this context, Li and coworkers have identified six novel genes in CNDT2.5 cells that may regulate cell growth and differentiation in normal and GEP-NET cells [13]. Of note, among these novel genes that OCT upregulated was *TGFBR2* encoding TGF-β type II receptor, a member of the Ser/Thr protein kinase family, which is part of the TGFB receptor superfamily. Apart from adding another facet of SST-TGF-β signaling crosstalk, i.e., mutual upregulation of components between the SST-SSTR and TGF-β systems, this would imply that TGFBR2 might be either involved in OCT indirect effects or being pivotal in opposing cell growth control via a downstream network.

Two of the most effective treatments for patients with NET (i.e., SSAs and peptide radionuclide therapy) depend on the expression of SSTR2 and SSTR5 on the tumor cell surface [1,4], and these SSTRs are crucial therapeutic targets for SSAs and peptide radioreceptor therapy in NET treatment. Assessing co-therapeutics’ effect on SSTR expression in NT-3 will help select effective drug combinations. Since a substantial fraction of NET tumors—especially panNET—do not express sufficient levels of SSTR2 and SSTR5, these patients are currently not amenable to effective treatments. Hence, pretreatment with TGF-β might thus be a promising approach to increase SST or SSTR2/5 levels and, thereby, sensitize panNETs to therapy with SSAs or peptide radionuclide therapy.

Our previous NGS-based identification of miRNAs expressed in BON, QGP, and NT-3 cells has yielded many miRNAs with potential involvement in panNET biology [10]. Here, we identified miR-221 regulated by OCT treatment in BON cells. Upregulation of this miRNA has been described in several types of human tumors and suggested to act as an oncogene or tumor suppressor, depending on the tumor type (reviewed in [18]). In human pancreatic cancer, upregulated miR-221/222 expression was associated with inhibited apoptosis and enhanced proliferation and invasion of the cancer cells [18,23]. Therefore, we validated a panel of cancer-associated genes by qPCR analysis and found TIMP2 mRNA levels to be increased in OCT-treated BON cells, an effect that is likely mediated through miR-221. However, whether TIMP2 is a direct functional target of miR-221 in BON and whether the OCT-induced TIMP2 upregulation results in reduced levels of the preferred TIMP2 substrate, matrix metalloproteinase 2 (MMP-2), and decreased invasion [18,24] needs to be validated in miR-221 mimic transfection experiments. Unfortunately, no data are available on the role of miR-221 in panNETs, but it may be of interest here that miR-221 has been shown to contribute to NE differentiation in androgen-independent prostate cancer [25].

Although we have not explicitly tested a possible regulatory function of TGF-β on the previously identified miRNAs here, it should be mentioned that miR-221 has been reported to interact with TGF-β1 signaling or activation in various ways [26,27,28,29,30]. Two other miRNAs regulated in BON in response to SST treatment (AP001273.1 and AC079951.1) still await functional characterization before their role in the SST-induced differentiation of BON can be discussed. The same applies to the miRNA AL449212.1 in SST-treated NT-3 cells. Only a comparatively low number of miRNAs have been identified here to be regulated by SST, or OCT does not appear to be caused by low SSTR expression. Otherwise, numbers would be expected to be higher in NT-3 cells. Some of the tumor-associated miRNAs that SST or the SSAs do not directly target might contribute to the increased proliferation index of BON and QGP compared to authentic panNET cell lines, such as NT-3.

Based on our in vitro findings, we can now—in a hypothesis-driven manner—investigate patient samples. Indeed, studying the effect of SSA treatment in tumor samples would require a dedicated study protocol to collect biopsies before and after treatment initiation, and we are aiming to perform this study in the future.

## 4. Materials and Methods

### 4.1. Cells

The BON cell line was established in 1991 as BON-1 from a lymph node metastasis of an insulinoma [5] and the QGP-1 line from the primary pancreatic carcinoma [6]. BON-1 cells were a kind gift from Dr. C.M. Townsend (University of Texas, Galveston, TX, USA). In contrast, QGP-1 cells were purchased from the JCRB Cell Bank/XenoTech (Cambridge, UK) and cultured in DMEM/Ham’s F12 (1:1) (BON-1), or RPMI 1640 (QGP-1), supplemented with 10% FBS and penicillin/streptomycin. The NT-3 cell line was established and characterized by us recently [9] and maintained under semi-adherent conditions in collagen IV-coated culture flasks in RPMI 1640 medium supplemented with 10% FCS, penicillin/streptomycin, HEPES, epidermal growth factor (EGF, 20 ng/mL), and fibroblast growth factor 2 (FGF2, 10 ng/mL) (Peprotech, Hamburg, Germany).

### 4.2. RNA Isolation and Quantitative Real-Time RT-PCR (qPCR)

RNA isolation and purification for qPCR were performed with PeqGold (PeqLab, Erlangen, Germany). Total RNA (2.5 μg) was reverse transcribed to cDNA using 2.5 μM random hexamers and 200 U M-MLV-Reverse Transcriptase. The cDNA samples were subjected to PCR on an I-Cycler (BioRad, Munich, Germany) with Maxima SYBR Green Mastermix (Thermo Fisher Scientific, Waltham, MA, USA). The threshold (C_t_) values of the genes-of-interest were normalized to those of the housekeeping gene, TATA box-binding protein (TBP), using the 2^−ΔΔCt^ method. TBP was chosen for normalization since it exhibits moderate-to-low expression (C_t_ values of ~25), being in the range of the C_t_ values (between 20 and 30) for most of the analyzed NE marker genes. The sequences of the primers used for qPCR can be found in [10]. The primers for TIMP2 (GenBank Accession: NM_003255) were 5′-AAG CGG TCA GTG AGA AGG AAG-3′ (forward) and 5′-GGG GCC GTG TAG ATA AAC TCT AT-3′ (reverse).

### 4.3. Immunoblot Analysis

Cell lysis and immunoblotting were carried out as described in detail elsewhere [10]. The following primary antibodies were used: CgA: Monoclonal Mouse Anti-Human Chromogranin A, clone DAK-A3, Dako, Glostrup, Denmark; SYP: Anti-Synaptophysin, Dako; SSTR2: Anti-Somatostatin Receptor 2 antibody [UMB1]-C-terminal #ab134152, Abcam, Cambridge, UK. Following washing and incubation with horseradish peroxidase-linked secondary antibodies, chemoluminescent detection of proteins was performed on a ChemiDoc XRS imaging system (BioRad) using Amersham ECL Prime Detection Reagent (GE Healthcare, Munich, Germany). The resulting signals were scanned densitometrically and normalized to those for the housekeeping genes, GAPDH or HSP90.

### 4.4. Cell Counting Assay

The proliferative activity of BON, QGP, and NT-3 cells was determined by cell counting. Assays were carried out under standardized culture conditions in either 12-well plates (BON, QGP) or 24-well plates (NT-3). A total of 50,000 cells were seeded and, after incubation for 72 h (BON, QGP), were treated separately with the following therapeutics or agents: SST, OCT, LAN (all from Merck, Darmstadt, Germany). In the case of TGF-β1 (premium grade, 5 µg/mL final concentration, Miltenyi Biotech, Bergisch Gladbach, Germany), NT-3 cells were treated for either 7 or 14 d. At the end of the incubation period, cells were detached by trypsinization, stained with trypan blue, and counted semi-automatically using a Cedex XS device and corresponding software (Roche Diagnostics, Mannheim, Germany) (BON, QGP) or manually using a Neubauer chamber (NT-3). As assessed by their failure to exclude the trypan blue dye, the number of dead cells was consistently below 1%.

### 4.5. Next-Generation Sequencing of miRNAs

The total RNA for NGS was isolated and purified with Qiagen miRNeasy Mini Kit (Qiagen, Hilden, Germany) and quality controlled on a TapeStation 4200 (Agilent Technologies, Santa Clara, CA, USA). RIN Scores were above 7 for all samples. Construction of sequencing libraries was performed with 200 ng of input total RNA using the NEXTFLEX^®^ Small RNA-Seq Kit v3 for Illumina^®^ Platforms (PerkinElmer, Waltham, MSA, USA) following the manufacturer’s protocol for a gel-free workflow. Quality control of the resulting libraries was performed on a TapeStation 4200 (Agilent Technologies, USA), showing a clean peak at approx. 160 base pairs (bp). Quantification was performed on a Qubit 2.0 fluorometer (Thermo Fisher Scientific). The two libraries (corresponding to BON and QGP cells) were multiplexed and sequenced on a HiSeq 4000 (Illumina, San Diego, CA, USA) lane with 50 bp single-read sequencing. Trimmed reads were mapped to GRCh38 using STAR aligner v2.7.2b and Gencode v24 annotations. Additionally, miRNA annotations from Gencode V29 were added (ENCODE reference ENCSR564GPK). Differentially expressed miRNAs were identified using DESeq2 (v1.36.0) [31].

### 4.6. Statistical Analysis

Statistical significance was calculated using the Student’s t-test, Mann–Whitney U test, or the Wilcoxon test when calculating means and standard deviations from three or more independent assays. Results were considered significant at *p* < 0.05, except for miRNA analysis, where differences between treatments were assumed significant at *p* < 0.1.

## 5. Conclusions

In this study, we have shown that NE marker expression in the panNET lines NT-3, BON, and QGP is subject to regulation by SST and SSAs, and these agents may thus potentially be able to alter the NE and well-differentiated phenotype in addition to their well-known effects on proliferation. Of note, we found for LAN, but not OCT or SST, an unexpected stimulatory rather than an inhibitory effect on BON proliferative activity. This extends earlier observations of different behaviors of BON and QGP cells. We also revealed that SST-SSTR2/5-TGF-β signaling crosstalk operates in NT-3 cells to control NE differentiation and cell proliferation. We conclude that the NT-3 and BON cell lines, despite their different maturity state, are principally relevant as NET models to study SSA therapeutic effects and are convinced that these data will further aid in the decision whether or not the use of any of these three cell lines will be helpful in future studies on NET cell models.

## Figures and Tables

**Figure 1 ijms-23-15868-f001:**
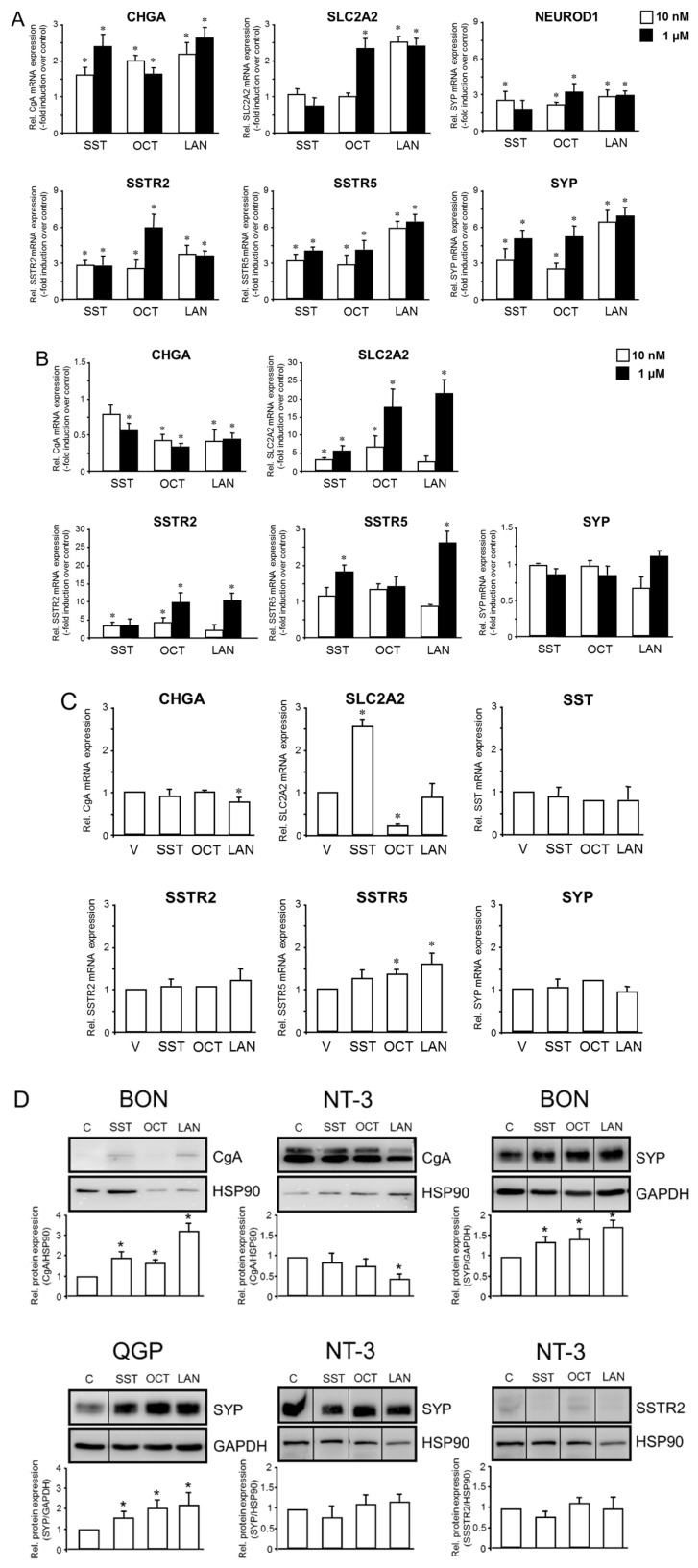
Response of various NE markers to treatment with SST or SSAs in BON, QGP, and NT-3 cells. (**A**) BON, (**B**) QGP, or (**C**) NT-3 cells were treated with the indicated agents (BON, QGP: 10 nM or 1 µM; NT-3: 1 µM) for 24 h followed by RNA isolation and qPCR for the indicated NE markers. Data are displayed as mean ± SD after normalization with the housekeeping gene TBP relative to untreated controls (set arbitrarily at 1.0) and are representative of three assays. The asterisks (*) indicate a significant difference (*p* < 0.05, Student’s *t*-test). Please note that the scaling on the ordinates may vary among the graphs. (**D**) BON, QGP, or NT-3 cells were stimulated with the indicated agents for 48 h, followed by lysis and immunoblotting for CgA, SYP, or SSTR2, and either HSP90 or GAPDH as a loading control. The graphs below the blots show results from the quantification of densitometric readings from three blots, each derived from a separate experiment (mean ± SD, n = 3). The asterisks (*) indicate a significant difference (*p* < 0.05, Wilcoxon test) relative to untreated controls set arbitrarily at 1.0. The vertical lines between the lanes of some blots denote the removal of irrelevant lanes.

**Figure 2 ijms-23-15868-f002:**
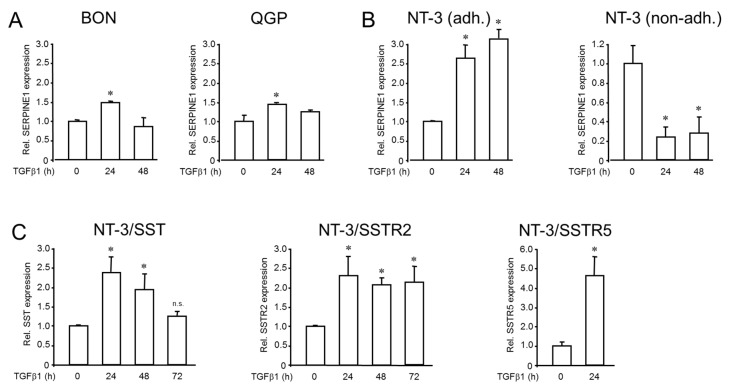
Effect of TGF-β1 treatment on expression of SERPINE1, SST, SSTR2, and SSTR5 in BON, QGP, and NT-3 cells. Cells were challenged with TGF-β1 (5 ng/mL) for the indicated times, followed by RNA isolation and qPCR analysis: (**A**) SERPINE1 in BON and QGP cells. (**B**) SERPINE1 in adherent (adh.) and non-adherent (non-adh.) NT-3 cells. Data in A-B represent the mean ± SD of four independent assays (*p* < 0.05, Wilcoxon test) relative to the untreated (0) control cell set at 1.0. (**C**) SST, SSTR2, and SSTR5 expression in NT-3 cells as measured by qPCR analysis. The data shown are from a representative assay out of three assays performed in total (mean ± SD after normalization to TBP and relative to untreated control cells set at 1.0). The asterisks (*) denote a significant difference (*p* < 0.05, Student’s *t*-test).

**Figure 3 ijms-23-15868-f003:**
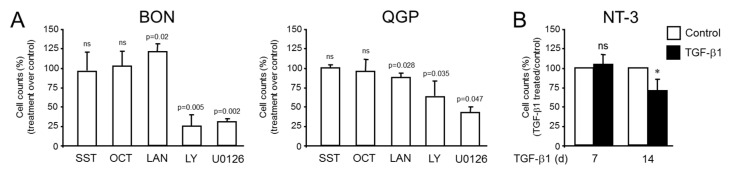
Effect of SST, SSAs, or TGF-β1 treatment on the proliferative activity of BON, QGP, and NT-3 cells. (**A**) BON and QGP cells were seeded at a density of 50.000 cells per 12-well on day 1 and treated on day 2 with either SST, OCT, or LAN (each at 1 µM) for 72 h in standard growth medium. Cells in parallel wells were incubated with LY294002 (LY, 50 µM) or U0126 (10 µM), both of which served as controls for growth-inhibitory agents. Data are plotted relative to vehicle-treated control cells set at 100% (not shown) and represent the mean ± SD of 3-5 independent assays (*p* < 0.05, Wilcoxon test) (**B**) NT-3 cells were challenged with TGF-β1 (5 ng/mL) for the indicated times (7 or 14 d), and cell counts were subsequently determined. Data shown are the mean ± SD relative to control cells (white-filled bar) set at 100% for both time points (n = 3). The asterisk (*) denotes a significant difference (*p* < 0.05, Wilcoxon test); ns, non-significant.

**Figure 4 ijms-23-15868-f004:**
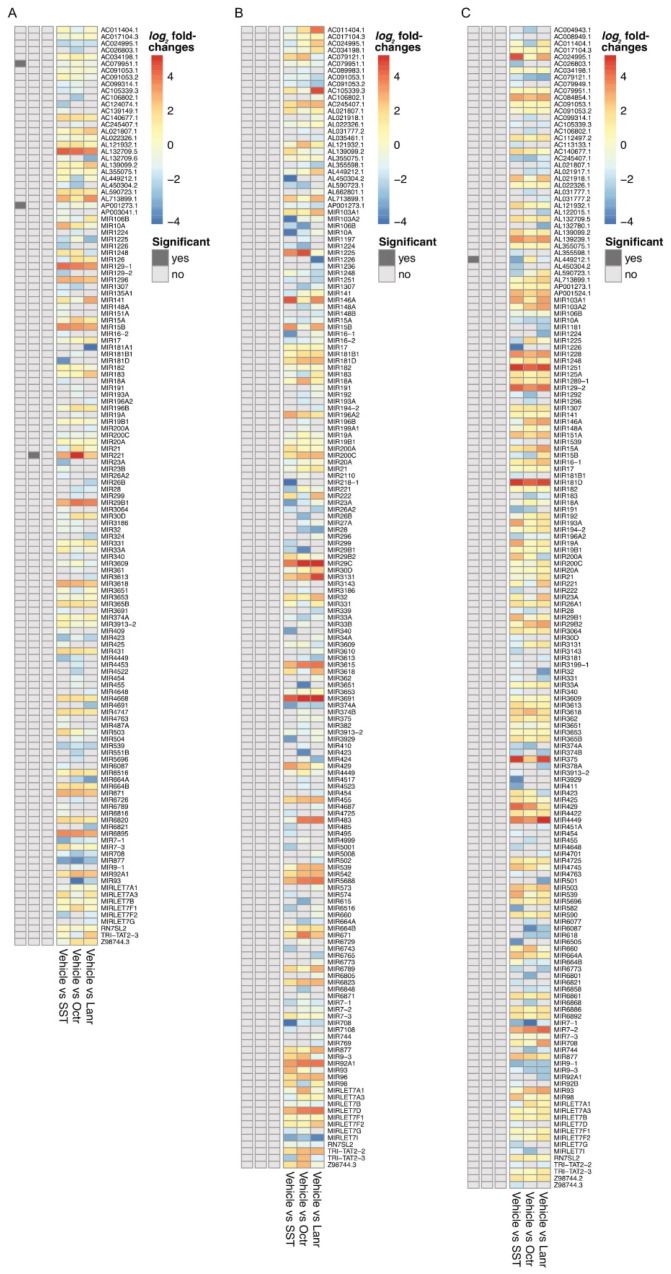
miRNA sequencing in BON (**A**), QGP (**B**), or NT-3 (**C**) cells treated with either vehicle, SST, OCT, or LAN. The heatmaps show log_2_ fold changes as estimated by DESeq2. Values above 0 (orange/red colors) denote miRNAs upregulated in the vehicle (no treatment) group compared to the treatment group. In contrast, values below 0 (blue colors) represent those upregulated miRNAs in the treatment group. Grey cells indicate that the miRNA was not found (missing values). The left columns show *p*-values below 0.1 (dark grey, considered significant) or above 0.1 (light grey, considered non-significant).

## Data Availability

All data are contained in the main text of this manuscript or in the Appendix A.

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
