# Peer review of "Differential Effects of Somatostatin, Octreotide, and Lanreotide on Neuroendocrine Differentiation and Proliferation in Established and Primary NET Cell Lines: Possible Crosstalk with TGF-β Signaling"

_ijms, 2022, doi:10.3390/ijms232415868_

Round 1

Reviewer 1 Report

Ungefroren et al. evaluated the differential effects of SST, OCT, LAN and TGF-β1 between pre-existing pancreatic NET cells and novel pancreatic NET cells NE-3.

The novelty of this paper is a little unclear.

The data presented in the referenced paper "Mol Cancer Res. 2018 Mar;16(3):496-507" clearly shows the superiority of NT-3 to BON and QGP.

In this paper, additional miRNA analysis was performed, but it only showed the obvious results that SST and OCT regulate miRNA. You should investigate whether the miRNA is also affecting patient samples and is related to treatment efficacy.

The figure should be easy to see. The characters in figure1 are illegible, and the lines in figures2, 3, and 4 are out of alignment. figure5 lacks ABC explanation.

You need at least the experimental data from mouse models.

We hope to see good results in the future.

Author Response

Dear Editor, dear Ms. Ilucz:

First of all, we would like to thank the reviewers for enthusiastic response to our manuscript. We appreciate their helpful comments, which we have addressed below point by point. We strongly believe that their comments have considerably enhanced the quality of our manuscript. All changes to the original text in the revised version have been highlighted in the “track changes” mode.

Reviewer 1

Ungefroren et al. evaluated the differential effects of SST, OCT, LAN and TGF-β1 between pre-existing pancreatic NET cells and novel pancreatic NET cells NE-3.

  1. The novelty of this paper is a little unclear.

Response: We have now included the novelty aspects of the manuscript even more clear and precise in the introduction and discussion.

In particular, we now stated more explicitly that

  1. the synthetic somatostatin and the two somatostatin analogues, octreotide and lanreotide, were for the first time systematically compared in the three available cell lines
  2. the so far underinvestigated signaling mode of these substances through TGF-b was studied
  3. the effects of octreotide, lanreotide and TGF-b on neuroendocrine marker expression and cell proliferation were analyzed
  4. the regulation of specific miRNAs by SST and SSAs was studied.

A deeper understanding of SSA action in NET tumor cells is of utmost important to further optimize this treatment strategy to prolong patient response to treatment. Having identified TGF-b as a potential regulator and effector of SSA response opens new windows for research to enhance SSA efficacy in patients.

  1. The data presented in the referenced paper "Mol Cancer Res. 2018 Mar;16(3):496-507" clearly shows the superiority of NT-3 to BON and QGP.

Response: That is true, however, we strived to compare cell lines since BON and GGP are still frequently used due to limited availability of the NT-3 cell line.

  1. In this paper, additional miRNA analysis was performed, but it only showed the obvious results that SST and OCT regulate miRNA. You should investigate whether the miRNA is also affecting patient samples and is related to treatment efficacy.

Response: This is an excellent idea, however, this study was designed as an in-vitro study. Unfortunately, we do not have a cohort of patients in our department that is large enough to study this. We will certainly aim at performing this study in the future; based on our in-vitro findings we can now in a hypothesis-driven manner investigate patient samples. Indeed, studying the effect of SSA treatment in tumor samples would require a dedicated study protocol to collect biopsies before and after treatment initiation. Without a valid hypothesis generated by in-vitro studies it will be impossible to obtain ethics approval for such a study. We have added a small paragraph on this to the end of the Discussion section.

  1. The figure should be easy to see. The characters in figure 1 are illegible, and the lines in figures 2, 3, and 4 are out of alignment. Figure 5 lacks ABC explanation.

Response: The characters and the lettering in Figure 1 were enlarged. The out-of-alignment of the lines in Figures 2, 3 and 4 (now 2, 3A and 3B) might have been caused by the conversion of the original word file into the pdf version, since in the original word version, which will be used for publication, characters are aligned and in an acceptable format. The ABC explanation has been added to Figure 4 (formerly Figure 5).

  1. You need at least the experimental data from mouse models.

Response: We agree with the reviewer, however, as mentioned above, this was an in-vitro study. Moreover, there are currently no valid mouse models for neuroendocrine neoplasms.

We hope to see good results in the future.

Additional changes made: As requested, we have rephrased the content of the manuscript that is highlighted in the plagiarism report. This primarily applied to the Material and Methods sections 5.1., 5.2., 5.3., and 5.4.

Reviewer 2 Report

The focus of this paper is to understand the differences in the response to STTR targeted therapies in three pancreatic neuroendocrine tumor cell lines namely Bon-1, QGP-1 and NT-3. The authors evaluate changes in gene expression and differences in IC50s to STTR targeted therapeutics OCT and LAN. Differences in cell behavior in response to TGF-B targeting is evaluated as well. A sequencing study is also performed to see key gene changes among the cell lines tested.

Specific comments:

Figure 3 and 4 can be merged. As it is Figure 4 alone does not add much

Figure 1 Western blots need to be repeated

Figure legends are not clear and better-quality figures should be provided

Abstract needs to be re-written and shortened. The reference from Leu has been mentioned twice and could be described in better way

Introduction has redundant information that can be moved in discussion

microRNA studies are not that revealing without further validation using PCR of their target genes

Author Response

Dear Editor, dear Ms. Ilucz:

First of all, we would like to thank the reviewers for enthusiastic response to our manuscript. We appreciate their helpful comments, which we have addressed below point by point. We strongly believe that their comments have considerably enhanced the quality of our manuscript. All changes to the original text in the revised version have been highlighted in the “track changes” mode.

Reviewer 2

The focus of this paper is to understand the differences in the response to STTR targeted therapies in three pancreatic neuroendocrine tumor cell lines namely Bon-1, QGP-1 and NT-3. The authors evaluate changes in gene expression and differences in IC50s to STTR targeted therapeutics OCT and LAN. Differences in cell behavior in response to TGF-B targeting is evaluated as well. A sequencing study is also performed to see key gene changes among the cell lines tested.

Specific comments:

  1. Figure 3 and 4 can be merged. As it is Figure 4 alone does not add much

Response: As requested, we have merged figures 3 and 4. As a consequence, the former Figure 3 has become Figure 3A, the former Figure 4 has become Figure 3B and the former Figure 5 has become Figure 4 in the revised version.

  1. Figure 1 Western blots need to be repeated

Response: We believe that the Western blots in Fig. 1 are all of good quality. It was only necessary to remove a few irrelevant lanes to enhance clarity. However, this did not compromise quantitation of band intensities and, therefore, the quantitative data shown in the graphs and calculated from three independent experiments truly reflect the differences among the various samples. Having said this, we would prefer to leave Figure 1 as it stands.

  1. Figure legends are not clear and better-quality figures should be provided

Response: Figure legends have been checked for clarity and in some cases slightly modified to enhance clarity. The bad quality of the figures is most likely due to the pdf conversion process, since in the original word version, which will be used for publication, the quality is acceptable.

  1. Abstract needs to be re-written and shortened. The reference from Leu has been mentioned twice and could be described in better way

Response: As requested, the abstract has been shortened and slightly modified to remove the Leu references.

  1. Introduction has redundant information that can be moved in discussion

Response: As requested, a redundant part from the last paragraph of the Introduction has been moved to the first paragraph of the Discussion section.

  1. microRNA studies are not that revealing without further validation using PCR of their target genes

Response: We have searched for target genes of miR-221 and have validated altered regulation of the miR-221/222 target gene, tissue inhibitor of metalloproteinase 2 (TIMP2), by qPCR analysis. The regulatory effect of OCT on TIMP2 is most likely a consequence of the OCT-induced decrease in miR-221 levels, however, this has to be confirmed in miR-221 mimic transfection experiments. These TIMP2 expression data have been included in the revised version of our manuscript as Figure S1.

Additional changes made: As requested, we have rephrased the content of the manuscript that is highlighted in the plagiarism report. This primarily applied to the Material and Methods sections 5.1., 5.2., 5.3., and 5.4.

Round 2

Reviewer 1 Report

Ungefroren et al. have corrected the text very well. The novelty of their research also made it easier for readers to understand.

In NT-3 cells, the crosstalk function between SST and TGF-β signaling was found to be more important than that of BON and QGP, and it was found to be important as PanNET cell model.

But I feel that some important results are missing.

Are there any results for 48 hours and 72 hours for NT-3/SSTR5 in Figure 2-C?

The figure should be easy to see. The lines in figures2, 3, and 4 are out of alignment.

Author Response

Ungefroren et al. have corrected the text very well. The novelty of their research also made it easier for readers to understand.

In NT-3 cells, the crosstalk function between SST and TGF-β signaling was found to be more important than that of BON and QGP, and it was found to be important as PanNET cell model.

But I feel that some important results are missing.

1. Are there any results for 48 hours and 72 hours for NT-3/SSTR5 in Figure 2-C?

Response: Yes, we had tested the effects of OCT and LAN on NT-3 cells also after 48 hours of stimulation (but not after 72 hours). The results of the 48 hour time point were quite similar to those of the 24 hour time point. Because of this and the fact that changes in mRNA levels normally occur within the first 24 hours of treatment, we decided not to display the 48 hour data in FIgure 2C. Since the Reviewer was just asking for this piece of information but did not specifically request us to include these data in Figure 2, we would like to leave Figure 2 as it is. However, in case that the Reviewer wishes to have these data included in the manuscript, we are happy to supply them in Figure 2C or, alternatively, in a Supplementary file.   

The figure should be easy to see. The lines in figures2, 3, and 4 are out of alignment.

Response: As stated in our response to the same comment in round 1 we believe that this only applies to the PDF version and may have been caused by the conversion process. I have checked the latest Word version attached here and all lines in Figures 2-4 appear to be aligned and okay.

Reviewer 2 Report

Most of the comments have been addressed.

Author Response

Most of the comments have been addressed.

We would like to thank this reviewer for his appreciation of our revision.